# Parental Microbiota Modulates Offspring Development, Body Mass and Fecundity in a Polyphagous Fruit Fly

**DOI:** 10.3390/microorganisms8091289

**Published:** 2020-08-24

**Authors:** Binh Nguyen, Anh Than, Hue Dinh, Juliano Morimoto, Fleur Ponton

**Affiliations:** 1Department of Biological Sciences, Macquarie University, Sydney, NSW 2113, Australia; thi-binh.nguyen@hdr.mq.edu.au (B.N.); anh.than-the@hdr.mq.edu.au (A.T.); dinhhue89@gmail.com (H.D.); juliano.morimoto@abdn.ac.uk (J.M.); 2Department of Entomology, Vietnam National University of Agriculture, Trau Quy, Gia Lam, Hanoi 100000, Vietnam; 3School of Biological Sciences, Zoology Building, Tillydrone Ave, Aberdeen AB24 2TZ, UK

**Keywords:** transgenerational effects, gut microbiota, offspring performance, life-history traits, reproductive success

## Abstract

The commensal microbiota is a key modulator of animal fitness, but little is known about the extent to which the parental microbiota influences fitness-related traits of future generations. We addressed this gap by manipulating the parental microbiota of a polyphagous fruit fly (*Bactrocera tryoni*) and measuring offspring developmental traits, body composition, and fecundity. We generated three parental microbiota treatments where parents had a microbiota that was non-manipulated (control), removed (axenic), or removed-and-reintroduced (reinoculation). We found that the percentage of egg hatching, of pupal production, and body weight of larvae and adult females were lower in offspring of axenic parents compared to that of non-axenic parents. The percentage of partially emerged adults was higher, and fecundity of adult females was lower in offspring of axenic parents relative to offspring of control and reinoculated parents. There was no significant effect of parental microbiota manipulation on offspring developmental time or lipid reserve. Our results reveal transgenerational effects of the parental commensal microbiota on different aspects of offspring life-history traits, thereby providing a better understanding of the long-lasting effects of host–microbe interactions.

## 1. Introduction

The experience of parents can influence the behaviour, performance, and fitness of future generations [1,2,3,4]. Parental effect is defined as any effect on offspring phenotype that is determined by the genotype or environmental experience of their parents [5,6]. Parental effects can be paternal and/or maternal and have been reported widely in plants [7], insects [8], and vertebrates [5]. Molecular mechanisms responsible for parental effects likely involve epigenetic modifications such as DNA methylation, chromatin modification, and noncoding RNA [9].

The commensal microbiota plays an important role in many physiological functions of its host [10,11,12], with evidence of transgenerational epigenetic implications on descendants [13,14,15,16,17]. In vertebrates, including humans, the microbiota has been detected in the placenta, amniotic fluid, and meconium, supporting the “in utero colonization hypothesis” which is crucial for the metabolic function, immune development, and further health parameters of neonates during placental development [18]. In insects and, in particular, in tephritid fruit fly species, adult females possess a symbiont-rich organ called the ovipositor diverticulum, which smears the egg surface with symbionts before the eggs are deposited [19]. Females of the Mediterranean fruit fly *Ceratitis capitata* not only deposit bacteria over the surface of freshly produced eggs but also provide eggs with lysozyme and antibacterial polypeptides which eliminate pathogens while facilitating the development of beneficial bacteria [20,21,22].

The microbiota and/or microbial products that offspring inherit from their parents can greatly impact offspring fitness, especially in the early life stages [15,16,17,23]. Research in mammals shows, for example, that signalling from maternal microbial molecules shapes the development and function of the neonatal immune system and the first colonization of the gut microbiota in early life is a critical window for the health and development of offspring [17,24]. In insects, bacteria or bacterial fragments transferred from mothers to eggs can mediate transgenerational immune priming, a phenomenon in which parents prepare their offspring to fight against pathogens that they encountered in their environment [15,23]. Notably, in *Drosophila melanogaster*, manipulations of the gut microbiota directly affect host mating and reproductive behaviour, with transgenerational consequences on offspring body mass [13]. Furthermore, in vertebrates, alterations of the maternal gut microbiota induced by factors such as diet, environmental toxins, or obesity status can influence the establishment of the microbial community and increase metabolic disorders in offspring as a consequence of developing in a detrimental intrauterine environment (see, for instance, [25,26] and [18]). The effects of parental microbiota are, therefore, important in shaping offspring phenotype and fitness, however, empirical studies targeting the transgenerational effects of host–microbe interactions on life-history traits of descendants are still limited.

Previous studies on the polyphagous fruit fly *Bactrocera tryoni* have shown that two yeast strains from the microbiota, belonging to the genera *Hanseniaspora* and *Pichia*, play an important role in development [27] and the microbiota inherited from parents is essential for maximizing pupal production [28]. In the present study, we manipulated the parental microbiota of *B. tryoni* and measured the effects of this manipulation on offspring developmental traits (e.g., developmental time, percentage of egg hatching, pupal production, and adult emergence), body weight and lipid storage of juveniles and adults, and adult fecundity. We generated parental control flies which had an intact commensal microbiota, parental axenic flies for which the commensal microbiota was eliminated, and reinoculated parental flies for which the commensal microbiota was eliminated then reintroduced. Because the axenic treatment did not selectively remove the microbes from the digestive tract, we will use the term “microbiota” manipulation instead of “gut microbiota” manipulation throughout the manuscript. We predicted negative effects of the parental microbiota removal on offspring developmental traits due to the lack of vertically transmitted microbiota (prediction 1). As commensals are key modulators of host metabolism [10,29,30], lacking commensal microbiota at the parental generation may result in offspring with lower body weight and lower body lipid reserve (prediction 2). If prediction 2 was confirmed and because there is a positive correlation between female body size and the number of eggs produced [31,32,33,34], we predicted that daughters of parents that host a commensal microbiota produced more eggs than that of axenic parents (prediction 3). The insight gained here into the transgenerational effects of commensal microbiota manipulation on offspring fitness-related traits gives us a better understanding of how animal–microbe interactions can have long-lasting influences on host life-history traits.

## 2. Materials and Methods

### 2.1. Fly Stock

A *B. tryoni* lab-adapted colony was established in 2015 and has been maintained for more than 25 generations in non-overlapping generations, whereby larvae were allowed to develop in an artificial gel-based diet [35] and adults fed on ad libitum hydrolysed yeast (cat. no., 02103304, MP Biomedicals), fine sugar (CSR^®^ White Sugar), and water. All fly stocks and experiments were maintained at 25 ± 0.5 °C, 65 ± 5% relative humidity, and 12:0.5:11:0.5 light/dusk/dark/dawn photoperiod.

### 2.2. Experimental Design and Statistical Analyses

#### 2.2.1. Fly Rearing

The experimental design is illustrated in Appendix A. The parental axenic treatment was generated by egg dechorionation as described in [36]. Briefly, eggs were collected for 2 h from the lab-adapted colony, then, dechorionated for 3 min in 0.5% bleach (Peerless JAL^®^), followed by one wash in 70% ethanol for 1 min and three washes in sterile Milli-Q water. One-hundred treated eggs (N = 10 replicates) were then transferred using a fine paintbrush onto 25 mL sterile gel-based diet (see recipe and suppliers in Appendix A) in a 90-mm petri dish (cat. no., S6014S10, Techno Plas). The diet was prepared aseptically by mixing irradiated (10 kilograys for 21 h) brewer’s yeast and nipagin with freshly autoclaved solution A (water, sugar, wheat germ oil, sodium benzoate, and agar) and B (water and citric acid) in a biosafety cabinet. Axenic eggs were then allowed to develop into axenic adults in sterile conditions with clean air provided by a PCR working station (Airclean^®^ system AC600).

The parental control treatment (N =10 replicates) was generated from the same batch of eggs as the axenic treatment and treated similarly, except that eggs were washed using water only. The parental reinoculated treatment (N = 10 replicates) was generated by recolonizing the dechorionated eggs with microbes harvested from untreated eggs following these steps: (i) 100 untreated eggs were crushed firmly in 50 µL sterile Milli-Q water for 2 min by a handheld pestle cordless motor (cat. no., Z359971, Sigma), (ii) the solution was pipetted onto a diet that contained 100 dechorionated eggs, and (iii) petri dishes containing reinoculated eggs were left open for 5 min for water evaporation. Manipulations of the parental control and parental reinoculated treatments until adulthood were conducted in a non-sterile environment using the same procedure as for the axenic treatments.

Adults from all parental treatments were provided ad libitum autoclaved water and food (10 kg for 21 h irradiated hydrolysed yeast and sugar). The microbial status was determined in parental axenic and control eggs, larvae, and adults by a culture-dependent method and PCR quantification (Appendix A). Axenic samples were confirmed to be free of germs (Appendix A).

Eggs from 15-day-old axenic, control, and reinoculated flies were then collected to generate the axenic, control, and reinoculation offspring (referred to as axenic, control, and reinoculation treatment, respectively, in Figure 1). Eggs were collected for 24 h from a parental group of 5 males and 5 females (10 groups were generated for each treatment, N total = 30). One-hundred eggs from each group were deposited on a piece of black filter paper (S = 15 cm^2^, cat. no., 104705, Macherey-Nagel) settled on the surface of 25 mL non-sterile gel-based diet [35]. The black filter paper was used to assess egg hatching status at 4-day post egg seeding and was then discarded. All manipulations for the axenic, control, and reinoculation treatments in the offspring generation were conducted in a non-sterile environment using non-sterile food. All data below were collected from the offspring generation.

#### 2.2.2. Developmental Performance

The percentage of eggs that hatched after 4 days (i.e., fertility, N total = 30 replicates, 10 replicates per treatment) was calculated as [(N hatched eggs /(N hatched eggs+N unhatched eggs)]*100. We fitted a generalized linear model (GLM) with a binomial error distribution and quasi extension to test for the effect of the treatment (i.e., parental microbiota manipulation) on the percentage of egg hatching. P values were obtained from F-statistics. Student–Newman–Keuls (SNK) post hoc tests with a significance of 0.05 were applied to identify sample means that were different.

To estimate the percentage of pupal production, petri dishes containing the larvae were transferred to a 1.125 L plastic container (cat. no., 136000, Décor Tellfresh^®^) that had a 20 cm diameter plastic mesh window on one side and had been filled with 20 mg of vermiculite one day before the larvae started jumping out of the larval diet. Pupae were sieved from vermiculite when all larvae had pupated (i.e., 4 days after the first jump) and the total number of pupae was recorded. The percentage of pupal production (N total = 30) was calculated as (N pupae /N hatched eggs)*100. A GLM with a binomial error distribution and quasi extension, followed by SNK post hoc tests, was fitted to the data to test for the effect of the treatment on the percentage of pupal production and to compare means between treatments.

Partially emerged flies were individuals that emerged with a portion of their body stuck in the puparium. Percentage of partially emerged flies (N total = 30 replicates) was calculated as (N partially emerged flies /N pupae)*100. Each replicate had 40 pupae (i.e., N pupae = 40). A Kruskal–Wallis test followed by a Dunn post hoc test was used to detect the significant effect of treatment and compare the mean ranks between treatments.

Developmental time (N total = 30, 10 replicates per treatment) was measured in days as (i) egg–larval duration: from depositing the parental eggs on diet until the first pupation was observed, and (ii) egg–pupal duration: from depositing the parental eggs on diet until the first emergence was recorded. Kruskal–Wallis tests followed by Dunn post hoc tests were used to test for the effect of the treatment on the different variables and compare the mean ranks whenever treatment was significant.

#### 2.2.3. Body Composition

The body wet weight of larvae and adults was measured individually using a precision weighing balance (Sartorius^®^ ME5 scale, d = 0.0001 g). Larvae (N total = 60, 20 individuals per treatment) were sampled post pupation (i.e., 3rd instar larva) to test for the effects of treatment on larval weight. Male and female adults (N total = 120, 20 males and 20 females per treatment) were collected at day 10 post emergence to test for the effects of treatment and sex on adult weight. A GLM model with a Gaussian distribution was fitted to test for the effect of treatment on larval weight, or the effect of treatment, sex, and their interaction on adult weight. Analyses were followed by SNK post hoc tests to compare means between treatments and sex.

We used the same individuals as above to measure the percentage of lipid reserve in larvae (N total = 60) and female and male adults (N total = 120) using a chloroform extraction method [37]. Briefly, samples were dried at 55 °C for 48 h and dry weight measured using a Sartorius^®^ ME5 scale. Body lipid reserves were then extracted in three 24 h changes of chloroform (cat. no., 650498, Sigma). Chloroform was then removed and evaporated by leaving the samples in a fume cupboard (Dynaflow, unit no. FC100316) for 24 h. Samples were then redried and reweighed as previously. Percentage of lipid reserve was calculated as $\frac{Body\;dry\;weight-Lipid\;extracted\;body\;dry\;weight}{Body\;dry\;weight}*100$. A GLM model with binomial error distribution and quasi extension was fitted to test for the effect of treatment on larval lipid reserve and treatment, sex, and their interaction on adult lipid reserve. Analyses were followed by the SNK post hoc tests to compare means between treatments and sex.

#### 2.2.4. Fecundity

We set up 1.125 L cages with a group of five females and five males (one-day-old, N total = 30 cages, 10 cages per treatment). Flies were provided ad libitum food (Hydrolysed yeast and sugar) and water. The bottom of a 35 mm diameter petri dish (cat. no., CLS430165, Corning^®^) was used as the egg collection device in each cage. The petri dish contained 2 mL of water flavoured by natural apple essence (Foodie flavours™, 1 mL L^−1^) and was covered by a thin layer of parafilm (M laboratory^®^) that has numerous perforations on the surface for females to insert their ovipositors and lay eggs. Egg collection started at day 14 post emergence for 10 days. The number of eggs produced per cage per day was counted and the average number of eggs produced per female per day was estimated as a proxy of fecundity. No female died during the egg collection period. To test for the effect of treatment on offspring fecundity, we fitted a GLM model with a Gaussian distribution, followed by SNK post hoc tests to compare means between treatments.

## 3. Results

### 3.1. Effects of Parental Microbiota Manipulation on Offspring Developmental Traits

We found a significant effect of treatment on the percentage of egg hatching (GLM: F_2,27_ = 8.579, *p* = 0.0013) with the percentage of egg hatching being about 17% and 13% higher in offspring of the control and reinoculation treatments, respectively, compared to that of axenic treatment (control: 88.2%, reinoculation: 84.5%, axenic: 71.7%, Figure 1A). Similarly, the treatment influenced pupal production (GLM: F_2,27_ = 5.124, *p* = 0.013). Offspring of the control treatment produced approximately 8% more pupae than that of axenic treatment (87.67% vs. 79.37%), meanwhile, the percentage of pupae produced by offspring of reinoculation treatment (82.38%) was around 3% more than that of the axenic treatment but was not significantly different from both the control and axenic treatments (Figure 1B).

Manipulation of parental microbiota affected the percentage of partially emerged adults (Kruskal–Wallis: χ^2^ = 6.521, df = 2, *p* = 0.038). Pupae from axenic offspring emerged significantly more as partially emerged adults compared to that of the control (1.01% versus 0.12%, Dunn test, p-adjusted = 0.05, Figure 1C) and reinoculation treatments (1.01% versus 0.16%, Dunn test, p-adjusted = 0.046, Figure 1C). The percentage of partially emerged adults was not different between the offspring of the control and reinoculation treatments (Dunn test, p-adjusted = 0.918, Figure 1C). The developmental time of the offspring was not affected by treatment, with egg–larval duration lasting on average 6.9 ± 0.02 days (Kruskal–Wallis: χ_2_ = 2.62, df = 2, *p* = 0.877) and egg–pupal duration 18.3 ± 0.04 days (Kruskal–Wallis: χ_2_ = 0.806, df = 2, *p* = 0.669).

### 3.2. Parental Microbiota Affects Offspring Body Weight but Not Lipid Reserves

Larval body weight of offspring was influenced by treatment (GLM: F_2,57_ = 4.685, *p* = 0.013), with larvae from the control treatment being about 1 mg heavier than that of axenic treatment; body weight of larvae from the reinoculation treatment was at intermediate (control: 15.35 ± 0.3 mg, axenic: 14.25 ± 0.19 mg, reinoculation: 14.65 ± 0.27 mg, Figure 1D). The adult body weight of the offspring was significantly impacted by treatment (GLM: F_2,117_ = 4.141, *p* = 0.018) and sex (GLM: F_1,116_ = 419.3, *p* < 0.001), however, the interaction between treatment and sex was marginally significant (GLM: F_2,114_ = 2.843, *p* = 0.062). Females of the reinoculation treatment (20.88 ± 0.3 mg) were approximately 0.5 and 1.2 mg heavier than females of the control and axenic treatments (20.33 ± 0.38 and 19.67 ± 0.31 mg, respectively, Figure 1E). In males, the trend was slightly different whereby males of the control treatment were significantly lighter than males of the reinoculation and axenic treatments (control: 14.45 ± 0.32 mg, reinoculation: 15.53 ± 0.24 mg, axenic: 15.26 ± 0.28 mg, Figure 1E).

The percentage of lipid reserve was, however, not affected by parental microbiota manipulation in both larvae (GLM: F_2,57_ = 0.32, *p* = 0.728, on average 29.1 ± 0.25%) and adults (GLM: F_2,117_ = 0.353, *p* = 0.703). Lipid reserve was higher in males (12.4%) than in females (9.84%, GLM, F_2,116_ = 44.46, *p* < 0.001) with no significant interaction between sex and treatment (GLM: F_2,116_ = 0.847, *p* = 0.431).

### 3.3. Parental Microbiota Increases Offspring Fecundity

Treatment influenced the number of eggs produced by offspring (GLM: F_2297_ = 3.703, *p* = 0.026) with females of the reinoculation and control treatments producing about 7 eggs per day more than females of the axenic treatment (reinoculation: 32 ± 2 eggs, control: 30 ± 2 eggs, axenic: 24 ± 3 eggs, Figure 1F). The number of harvested eggs varied with time (GLM: F_9288_ = 5.62, *p* < 0.001) but there was no significant interaction between treatment and time (GLM: F_18,270_ = 0.639, *p* = 0.867). Overall, the number of eggs per day fluctuated from day 14 to day 20 before reaching a peak of 48 ± 4 eggs at day 21 post emergence; after that, egg number decreased gradually to 32 ± 2 eggs per day (Appendix A).

## 4. Discussion

In this study, the manipulation of parental microbiota had multiple effects on offspring performance. The results confirmed prediction 1 whereby microbiota-deficient parents generated offspring with a lower percentage of egg hatching, lower pupal production, and a higher number of partially emerged adults. We also found evidence of a lighter body weight in larvae and female offspring of axenic treatment, which partly confirmed our prediction 2, however, lipid reserve was not significantly different across offspring of axenic and non-axenic treatments. Reinoculating axenic parents with a microbiota harvested from control parents restored, however, some traits in offspring that were negatively affected by microbiota depletion in parents. Finally, following prediction 3, the fecundity of daughters from axenic parents was lower than that of control and reinoculated parents. It is important to emphasize here that while the treatment used to manipulate the microbiota may have caused nonspecific effects on some traits of the host, previous studies have confirmed that the main effect on insects’ performance is due to the absence of microbiota [28,38,39,40]. In our study, the egg dechorionation method did not interfere with the egg hatching success or the developmental time of parents (Appendix A). In addition, the developmental traits measured for the offspring of our reinoculation treatment were not different than that of the control treatment. Thus, the lower developmental and fitness-related traits of the axenic offspring are likely linked to the deficiency in parental microbiota rather than any side effects caused by the egg treatment method.

Our study showed that the offspring of axenic parents had a substantially lower percentage of hatching than that of non-axenic parents, resulting in a lower number of pupae produced. This finding is in agreement with previous research in the pine weevil, showing that the hatching success of eggs laid by mothers with a native microbiota was significantly higher than that of axenic mothers [41]. While the mechanism is unclear, it is possible that the presence or absence of different microbes on the egg surface and in the egg environment (i.e., in the diet in our case) regulates egg hatching as shown in previous studies in insects and nematodes [42,43,44]. For instance, the close physical contact between common bacteria in the host’s intestine and the eggs of parasitic nematodes has been confirmed to regulate egg hatching, with incubations with different bacteria leading to different egg hatching percentages. In addition, reducing the number of bacterial contacts with eggs can significantly decrease hatching percentage [43,44]. In our experiment, we manipulated the microbiota of parents and eggs deposited by axenic and non-axenic parents that were exposed to the same non-sterile diet. However, because eggs did not harbor the same microbial community at the beginning of the experiment (i.e., eggs delivered by axenic parents lack vertically transmitted microbes), the microbiota growing in the diet after egg seeding might had been different [16], thus, likely affecting egg hatching status. This experiment did not allow us to explore the mechanisms involved in the lower hatching percentage of eggs delivered by axenic parents and more investigations are needed. We also found a higher percentage of partially emerged adults in offspring of axenic parents, which, to our knowledge, has never been reported previously. This might be explained by the difference in metabolic status between axenic and non-axenic parents and the resource investment of females on eggs before seeding. Axenic individuals might lack essential nutrients that can be provided by beneficial microbes [12,45], hence, possibly having a lower investment on eggs, which resulted in a long-term implication on offspring quality [46,47,48]. This is particularly the case for oviparous insects where maternal investment is fixed at the time of egg laying, directly shaping early development with fitness consequences to the offspring [48,49].

Our axenic treatment, where the parental microbiota was removed, resulted in lighter offspring larvae and adult females. This supports previous findings in *D. melanogaster* whereby mating pairs that host different gut bacterial species generate daughters (but not sons) with different body weights [13]. Interestingly, sex-specific effects on body traits have also been observed in *Drosophila* at the parental generation when the microbiota is removed [40,50]. The mechanisms responsible for these effects are still debated, but some hypotheses argue that females’ high energy demand for egg production and the physiological differences between males and females (and thus, their microbiota composition) might explain this [51,52,53]. It was also surprising in our data that, while parental microbiota manipulation substantially influenced offspring body weight, no significant effect on body fat reserve was found. This result is different to what has been reported before in fruit flies and mosquitoes, at least for the first generation, indicating that the metabolic effect of axenic treatment on lipid metabolism might vary and may be lost at the offspring generation [40,54].

Manipulation of the microbiota at the parent generation also influenced offspring egg productivity with offspring of control and reinoculated parents producing more eggs than that of axenic parents. In insects and other arthropods, it is well established that parental endosymbionts affect offspring reproduction (reviewed by [55]). However, to our knowledge, this is the first time effects of the microbiota on offspring fecundity are reported in insects, though it has been shown to substantially affect the parental reproductive output of many species, including fruit flies [14,56,57], mosquito [54,58], and bean bug [59]. Given that insect fecundity is generally positively correlated with female body size [31,32,33], the higher number of eggs produced by the offspring of reinoculated parents compared with that of axenic parents might be linked to the greater body weight of females from reinoculated parents. However, because offspring of control parents (that body weight was intermediate) also produced more eggs than offspring of axenic parents and a similar number of eggs than offspring of reinoculated parents, fecundity may also be impacted by other factors such as the association with some specific bacterial species [14,60]. The gut bacterium symbiont *Acetobacter* (but not *Lactobacillus*) in *Drosophila*, for example, has been shown to regulate the activity of enzymes that convert aldehydes to carboxylic acids in the ovaries, which largely affects host oogenesis [14].

In conclusion, our study provides insight into the transgenerational effects of the commensal microbiota in a polyphagous fruit fly, showing that the elimination of the microbial community in parents could lead to fitness consequences in the next generation. Interestingly, when eggs from axenic parents were reinoculated with microbes, hatchings recovered some traits of offspring from unmanipulated parents, underlining the importance of the microbes that are directly transmitted by the mother to its eggs. Overall, the effects of parental microbiota manipulation on offspring reproduction described here extend our understanding of the long-lasting fitness implications of host–microbe interactions to both present and future generations, highlighting the potential evolutionary links between the host and its microbiota, which can drive evolutionary adaptations.

## Figures and Tables

**Figure 1 microorganisms-08-01289-f001:**
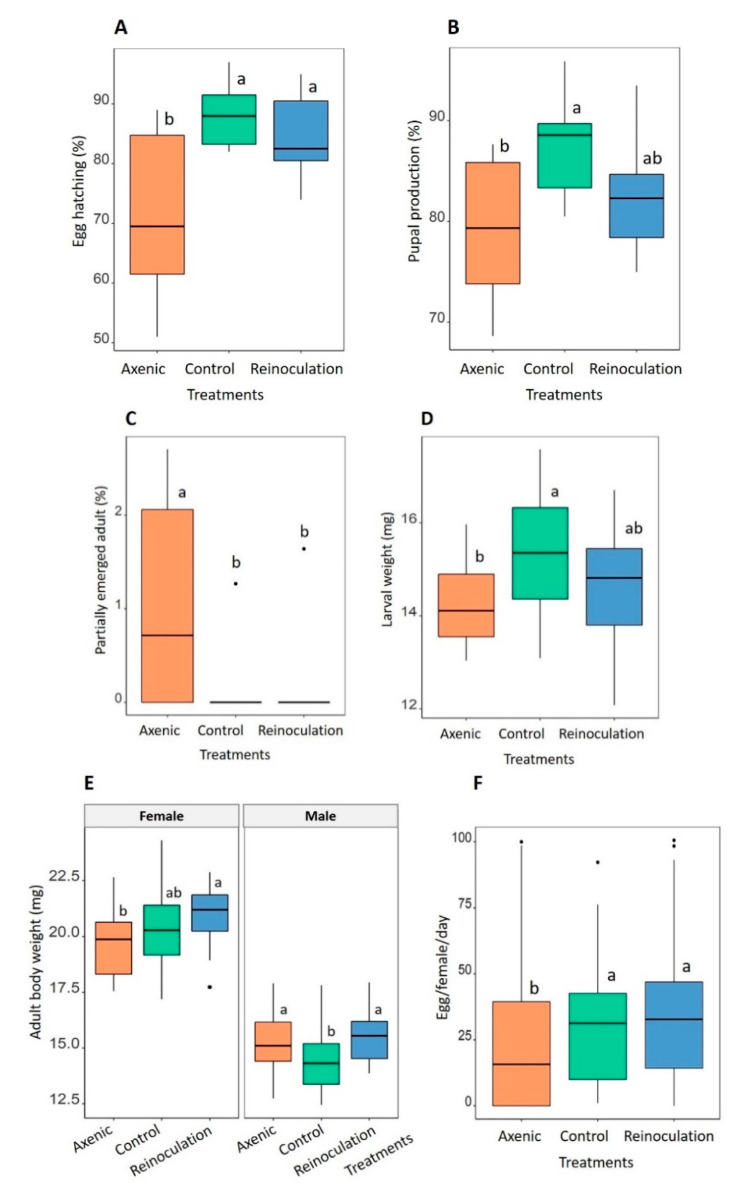
Effect of parental microbiota manipulation on (**A**) percentage of egg hatching, (**B**) pupal production, (**C**) partially emerged adults, (**D**) body weight of larva, (**E**) body weight of adult, and (**F**) female fecundity. Different letters indicate a significant difference between the treatments ((**A**,**B**,**D**–**F**) SNK post hoc test and (**C**) Dunn post hoc test, *p* < 0.05.

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
