# Peer review of "Parental Microbiota Modulates Offspring Development, Body Mass and Fecundity in a Polyphagous Fruit Fly"

_microorganisms, 2020, doi:10.3390/microorganisms8091289_

Round 1

Reviewer 1 Report

This is a highly original and overall well written study. Authors tested the hypothesis that vertically transmitted microbes affect subsequent development and fecundity of offspring in the Queensland fruit fly. The entomological experiments are very well designed and the statistical analysis is rigorous. The conclusion that there are trans-generational effects of parental commensal microbiota on different aspects of offspring life-history traits is fully supported by the experimental evidence.

Sadly, however, the manuscript lacks microbiological information. Such information is crucial to complement the entomological findings. The data presented in the supplementary information is difficult to understand. It is not clear from supplementary Table 3 what is measured, and some of the information is puzzling. For example- the fifth line of the table (adult Male control) has 150 in the CFU column, but 0 in the 16S and ITS columns.

Furthermore, the parental control for eggs, from which all the offspring used in the experiment stem, shows 1 CFU, and 0.04 ITS (presumably the value generated by qpcr?). Did all the biological differences found in subsequent generations stem from the absence of this 1 CFU? What bacteria were present there?

Beyond clarifying the information in this table, it would be useful to show microbial data for the eggs and adults of the next generations, and to attempt to characterize their microbiome and compare it to the parental microbiome. If authors can provide these details, the manuscript would be suitable for “microorganisms”. Possibly, it may be difficult for them to go back and do such an analysis. In that case they should consider publication in a journal with an entomological focus.  

Author Response

Reviewer 1

Rev1: This is a highly original and overall well written study. Authors tested the hypothesis that vertically transmitted microbes affect subsequent development and fecundity of offspring in the Queensland fruit fly. The entomological experiments are very well designed and the statistical analysis is rigorous. The conclusion that there are trans-generational effects of parental commensal microbiota on different aspects of offspring life-history traits is fully supported by the experimental evidence.

Ans: We thank Reviewer 1 for the very positive comments on the originality and the quality of the manuscript.

We answered the reviewer’s comments in details.

Rev1: The data presented in the supplementary information is difficult to understand. It is not clear from supplementary Table 3 what is measured, and some of the information is puzzling. For example- the fifth line of the table (adult Male control) has 150 in the CFU column, but 0 in the 16S and ITS columns.

Ans: We thank the reviewer for the comment since we have done a mistake in supplementary Table 3 with the PCR quantification data of the control treatment being under the axenic treatment rows and vice versa. We fixed this and changed Supplementary Table 3 for clarity. We now have 2 Supp tables: Table S3 shows the total CFU count per sample for each culture medium (see supplementary document, line 47-50) and Table S4 shows the 16S rRNA and ITS concentrations (see supplementary document, line 52-54).

We updated the CFU calculation method in the supplementary document, information 1.

Lines 23-24: “The number of colony-forming unit (CFU) on LB, MRS, PDA dishes was measured. The total CFU number per sample was then averaged between the 3 replicates for each medium (see Table S3).”

Lines 32-33: “16S rRNA and ITS concentrations (ng/µL) were then averaged between the 3 replicates (see Table S4).”

We also updated these changes in the main text.

Line 115: Changed to “(Supplementary document, Tables S3 & 4)”

Line 320-323: Names of supplementary tables have been updated.

Rev 1: Beyond clarifying the information in this table, it would be useful to show microbial data for the eggs and adults of the next generations, and to attempt to characterize their microbiome and compare it to the parental microbiome. If authors can provide these details, the manuscript would be suitable for “microorganisms”. Possibly, it may be difficult for them to go back and do such an analysis. In that case they should consider publication in a journal with an entomological focus.  

Ans: The aim of our study was to investigate the developmental and reproductive effects of the vertically transmitted microbiota (i.e., parents to offspring). Note that our focus is on the microbiota as a ‘trait’, irrespective of its fine-grain composition. A longitudinal study of the microbiome over generations with taxonomic characterization, as suggested by the reviewer, is an interesting idea for future studies. However, we believe it is beyond the scope of this paper, largely because it would change the focus of our study. A detailed taxonomic identification is useful to understand the microbial dynamics, but would only provide a finer resolution to our findings, without changing the main message of our results.

Reviewer 2 Report

 Parental microbiota modulates offspring development, 2 body mass and fecundity in a polyphagous fruit fly

This study examines the effect of parental microbiota on the development of the offspring in fruitflies.  It is a useful and informative study given the increasing interest in this area

Please check throughout manuscript for typographical and grammar issues.

Title:

Parental ‘gut’ microbiota… might help if correct.

Introduction

Line 65-68. Perhaps this sentence…”We generated……re-introduced” should go after the next sentence. Also it is not clear how this sentence connects with the sentences before or after,

Line 62-80. I assume the commensal bacteria is from the gut? If so, please mention here. If any further information such as taxonomical/kingdom, etc, then that can be included before this paragraph. It appears to bacteria and fungi, based on the supplementary material

Materials and Methods

The source of a required material follows an unusual format e.g. Techno Plas, 95 cat no. S6014S10. Suggest using the format of cat. no., followed by company, city, country.

Line 111. Ad libitum in italics

Line 281. Mating

Line 296. Extracellular, while it can be assumed as not been proved in this study. All we know is that they are parental.

References. Please check your references.

Line 362. This reference is incomplete.

Author Response

Reviewer 2

Rev2: This study examines the effect of parental microbiota on the development of the offspring in fruitflies.  It is a useful and informative study given the increasing interest in this area

Ans: We thank the reviewer for the positive comments on our manuscript.

We answered the reviewer’s comments in details.

Rev2: Please check throughout manuscript for typographical and grammar issues.

Ans: We have now corrected this in the main text.

Rev2: Line 65-68. Perhaps this sentence…”We generated……re-introduced” should go after the next sentence. Also it is not clear how this sentence connects with the sentences before or after

Ans: We reorganised the paragraph has suggested by the reviewer.

Lines 62-67: “Previous studies on the polyphagous fruit fly Bactrocera tryoni have shown that the commensal microbiota plays an important role on development [27] and the microbiota inherited from parents is essential for maximizing pupal production [28]. In the present study, we manipulated the parental microbiota of B. tryoni and measured the effects of this manipulation on offspring developmental traits (e.g., developmental time, percentage of egg hatching, pupal production and adult emergence) […]”

Rev2: TITLE: Parental ‘gut’ microbiota… might help if correct.

Ans: Our protocol could not selectively manipulate the gut microbiota without any potential effects on other microbiota communities (e.g., surface microbiota). We therefore prefer to maintain the original title as it is more accurate to our experimental design. We clarify this in the main text:

See lines 71-73: “Because the axenic treatment did not selectively remove the microbes from the digestive tract, we will use the term “microbiota” manipulation and not “gut microbiota” manipulation throughout the manuscript.”

Rev2: Line 62-80. I assume the commensal bacteria is from the gut? If so, please mention here. If any further information such as taxonomical/kingdom, etc, then that can be included before this paragraph. It appears to bacteria and fungi, based on the supplementary material

Ans: One of the two studies cited here had some details about the taxonomy of the microbes they studied. We added this information in the text.

See lines 62-65: “Previous studies on the polyphagous fruit fly Bactrocera tryoni have shown that two yeast strains from the microbiota, belonging to the genera Hanseniaspora and Pichia, play an important role on development [27] and the microbiota inherited from parents is essential for maximizing pupal production [28].”

Minor comments

Rev2: The source of a required material follows an unusual format e.g. Techno Plas, 95 cat no. S6014S10. Suggest using the format of cat. no., followed by company, city, country.

Ans: Done

Line 111. Ad libitum in italics

Ans: Done

Line 281. Mating

Ans: Done

Line 296. Extracellular, while it can be assumed as not been proved in this study. All we know is that they are parental.

Ans: Following the reviewer’s comment, we removed the word “extracellular”.

See lines 292-295: “However, to our knowledge, this is the first time effects of the microbiota on offspring fecundity is reported in insects though it has been shown to substantially affect parental reproductive output of many species, including fruit flies [14,56,57], mosquito [54,58], and bean bug [59]."

References. Please check your references. Line 362. This reference is incomplete.

Ans: Done

Round 2

Reviewer 1 Report

The authors addressed the 2 major issues that i brought up - the problems with the supplementary table and the general question of microbial data - in a very satisfactory manner.

The tables are fine now, and strengthen the whole manuscript.

The authors make a very strong case for treating the microbiota as a "trait". They convinced me, so i see no obstacle to publishing the revision in "microorganisms".